# Vitamin D and Type 2 Diabetes Mellitus: Molecular Mechanisms and Clinical Implications—A Narrative Review

**DOI:** 10.3390/ijms26052153

**Published:** 2025-02-27

**Authors:** Héctor Fuentes-Barría, Raúl Aguilera-Eguía, Cherie Flores-Fernández, Lissé Angarita-Davila, Diana Rojas-Gómez, Miguel Alarcón-Rivera, Olga López-Soto, Juan Maureira-Sánchez

**Affiliations:** 1Vicerrectoría de Investigación e Innovación, Universidad Arturo Prat, Iquique 1100000, Chile; hefuentes_@unap.cl; 2Escuela de Ondontología, Facultad de Odontología, Universidad Andres Bello, Concepción 3349001, Chile; 3Departamento de Salud Pública, Facultad de Medicina, Universidad Católica de la Santísima Concepción, Concepción 3349001, Chile; raguilerae@ucsc.cl; 4Departamento de Gestión de la Información, Universidad Tecnológica Metropolitana, Santiago 7550000, Chile; cflores@utem.cl; 5Escuela de Nutrición y Dietética, Facultad de Medicina, Universidad Andres Bello, Concepción 3349001, Chile; 6Escuela de Nutrición y Dietética, Facultad de Medicina, Universidad Andres Bello, Santiago 7550000, Chile; diana.rojas@unab.cl; 7Escuela de Ciencias del Deporte y Actividad Física, Facultad de Salud, Universidad Santo Tomás, Talca 3460000, Chile; mrivera3@santotomas.cl; 8Facultad de Medicina, Universidad Católica del Maule, Talca 3460000, Chile; 9Facultad de Salud, Universidad Autónoma de Manizales, Manizales 170017, Colombia; sonrie@autonoma.edu.co; 10Facultad de Educación, Universidad Central de Chile, Santiago 7550000, Chile; juan.maureira74@gmail.com

**Keywords:** vitamin D, type 2 diabetes mellitus, molecular biology

## Abstract

Vitamin D has been widely studied for its implications on type 2 diabetes mellitus, a chronic condition characterized by insulin resistance, inflammation, and metabolic dysfunction. This review explores the molecular mechanisms underpinning vitamin D’s effects on glucose metabolism, inflammation, and adipogenesis, while assessing its potential clinical applications in type 2 diabetes. In its 1,25-dihydroxyvitamin D3 form, vitamin D modulates various metabolic processes, affecting proinflammatory cytokines and activating the AMPK pathway, inhibiting mTOR signaling, and promoting adipocyte differentiation. These effects enhance insulin sensitivity and reduce chronic inflammation, key contributors to metabolic dysfunction. In this context, the progression of prediabetes has been linked to vitamin D, which limits pathological progression and increases the likelihood of restoring a normal metabolic state, crucial in diabetes progression. Moreover, vitamin D has been reported to reduce the likelihood of developing diabetes by 15%, particularly in doses higher than the traditional recommendations for bone health. Despite promising evidence, discrepancies in study designs, serum vitamin D measurements, and population-specific factors highlight the need for standardized methodologies and personalized approaches. In conclusion, vitamin D has complementary therapeutic potential in treating type 2 diabetes, revealing gaps in research, such as optimal dosing and long-term effects across populations. Future studies should integrate molecular insights into clinical practice to optimize vitamin D’s impact on metabolic health.

## 1. Introduction

Vitamin D is considered a key link for human health due to its known bone and pleiotropic effects that can modulate chronic non-communicable diseases such as type 2 diabetes mellitus. While its classical actions related to bone metabolism are well-documented, its influence as a metabolic and molecular regulator presents new therapeutic opportunities. However, the specific mechanisms by which vitamin D affects insulin sensitivity, glucose metabolism, and adipogenesis remain incompletely understood [1,2].

Type 2 diabetes presents multiple clinical manifestations, among which the state of chronic hyperglycemia generated by insulin resistance and dysfunction of pancreatic β cells stands out. Its high prevalence is linked to factors such as unhealthy dietary patterns, sedentary lifestyles, and an aging population [3,4]. In this context, it has been observed that the progression of prediabetes is related to vitamin D and its limiting effect on the pathological progression process, where vitamin D has shown potential to increase the likelihood of restoring a normal metabolic state, a crucial factor in the progression of diabetes [5]. Furthermore, it has been reported that vitamin D can reduce the likelihood of developing diabetes by 15%, especially in doses higher than the traditional recommendations for obtaining bone benefits [6]. Similarly, adequate serum levels of vitamin D have been linked to a lower risk of type 2 diabetes and, therefore, a potential benign metabolic effect [7,8]. Nevertheless, findings in this area have been inconsistent, raising questions about dose–response interactions, individual variability, and the long-term efficacy of vitamin D supplementation [9,10]. Studies have shown that although supplementation appears to improve certain metabolic parameters, these effects are not uniform, suggesting the influence of underlying factors such as genetic predisposition, baseline vitamin D levels, and preexisting conditions [11].

Recently, vitamin D’s role in the regulation of inflammation and glucose metabolism at the molecular level has drawn increasing attention. Vitamin D in its active form, called 1,25-dihydroxyvitamin D3, after reacting with its receptor (VDR) can decrease multiple proinflammatory markers such as interleukin-6 (IL-6), monocyte chemoattractant protein-1 (MCP-1) and tumor necrosis factor alpha (TNF-α) by inhibiting the action of the NF-κB pathway [12,13]. Additionally, its role in modulating microRNA expression and inflammatory responses underscores its potential as a multifaceted metabolic regulator [14,15,16,17]. However, the translation of these molecular changes into clinical benefits remains uncertain due to heterogeneity in vitamin D deficiency cases across different populations and contexts.

A critical aspect of vitamin D’s regulatory role lies in its influence on adipogenesis. Its ability to modulate stem cell differentiation promotes the development of healthier adipocytes, suggesting a novel pathway for improving metabolic health [14,15]. Furthermore, interactions with processes such as autophagy and cellular homeostasis offer new perspectives on preserving cellular integrity. Notably, heat shock proteins (HSPs), such as HSP60 and HSP70, undergo significant changes in response to vitamin D modulation, although their specific roles in type 2 diabetes and their interactions with other metabolic pathways, such as lipogenesis and oxidative stress, remain incompletely characterized [18,19,20,21].

Currently, more than half of the world’s population has serum levels below 30 nmol/L of 25(OH)D, reflecting widespread deficiency attributed to factors such as adiposity, limited dietary intake, and inadequate sunlight exposure. Socioeconomic disparities and geographical variations further complicate the attainment of optimal vitamin D levels on a population scale [2,22,23].

Although supplementation strategies have been proposed, they face obstacles such as low adherence, variability in dosing protocols, and inconsistent outcomes. Against this backdrop, this review aims to explore the molecular mechanisms underpinning vitamin D’s regulatory effects on glucose metabolism, inflammation, and adipogenesis, while evaluating its potential clinical applications in type 2 diabetes.

## 2. Diabetes

Type 2 diabetes mellitus is a group of metabolic disorders that, unlike type 1 diabetes mellitus, are not of autoimmune origin [24,25,26,27,28,29,30,31,32,33,34,35]. Type 2 diabetes presents a state of hyperglycemia, the acute clinical manifestations of which include symptoms such as polyuria, polydipsia, fatigue and unexplained weight loss which, if left untreated, can progress to severe states of ketoacidosis and high risk of coma [24,25,26]. In the long term, this pathology can generate organic dysfunctions that mainly affect the visual, renal, nervous and cardiovascular systems, as well as the potential development of melanomas [27,28,29,30,31].

In this context, hyperglycemia is attributed to a combination of relative deficiency in the secretion of insulin by pancreatic β-cells and insulin resistance in peripheral tissues, mediated by multiple factors such as lipid accumulation in muscle and liver cells, chronic inflammatory state, and alterations in intracellular signaling pathways. This leads to a silent progression that only manifests evident symptoms once significant organic damage has occurred [24,25,26,27,28,29,30,31]. Similarly, the environmental factor is one of the primary modulators of the risk and progression of type 2 diabetes, where the presence of an obesogenic environment—characterized by excessive accumulation of visceral adipose tissue—exacerbates insulin resistance. This is due to increased release of free fatty acids, combined with dysregulation of adipokines (leptin and adiponectin), both of which contribute to the development of hyperglycemia and predispose individuals to a chronic inflammatory state, leading to degenerative effects on the heart muscle, eyes, kidneys, and nervous system [24,25,26,27,28,29,30,31]

From a molecular perspective, insulin resistance is mediated by alterations in signaling pathways such as PI3K/Akt and the translocation of glucose transporters (GLUT4). Furthermore, continuous overstimulation of insulin secretion by the pancreas leads to gradual β-cell exhaustion, amplifying insulin deficiency. These chronic metabolic and molecular processes, together with systemic inflammation, contribute to irreversible organ damage and the progression of complications associated with metabolic and molecular dysfunction [24,25,26,27,28,29,30,31].

The prevention and management of diabetes have traditionally focused on lifestyle changes, where a healthy diet and regular physical activity serve as the cornerstones for maintaining optimal body weight. In this sense, medical nutrition therapy (MNT) may be linked to early management of the condition, although its role in diagnosis is unclear. When combined with lifestyle modifications, it allows for the regulation of energy intake, which should include carbohydrates (45% to 65% of total energy intake), fats (≤25% of total energy intake), and proteins (1 to 1.2 g/kg of ideal body weight) [36,37,38,39].

Table 1 provides a comprehensive overview of type 2 diabetes mellitus, highlighting its characteristics, underlying causes, and long-term complications. It outlines the key clinical manifestations, including symptoms such as polyuria, polydipsia, and fatigue, which can progress to severe complications like ketoacidosis and organ dysfunction.

## 3. Vitamin D

Vitamin D is a fat-soluble vitamin that exists in two primary forms in the human body: ergocalciferol (vitamin D2), which is synthesized by plants, and cholecalciferol (vitamin D3), produced in the skin upon exposure to ultraviolet radiation. The synthesis of vitamin D3 is heavily dependent on adequate sunlight exposure, making it an essential nutrient that relies on environmental factors to reach optimal levels in the body [47,48].

Once absorbed into circulation, vitamins D3 and D2 undergo metabolic processes in the liver. The enzyme vitamin D-25-hydroxylase (CYP2R1) converts these forms into 25-hydroxyvitamin D, also known as calcifediol [25(OH)D]. This intermediate form, 25(OH)D, is further transformed in the kidneys by the enzyme 25-hydroxyvitamin D-1α-hydroxylase (CYP27B1) into the active, biologically available form—1,25-dihydroxyvitamin D, or calcitriol (CT) [1,25(OH)_2_D] [49].

Once formed, 1,25(OH)_2_D performs essential functions by binding to the vitamin D receptor (VDR) located in the cytoplasm of cells. This interaction initiates the formation of a VDR-RXR hormone complex, driven by the heterodimerization of VDR with the retinoid X receptor [50]. Within the nucleus, 1,25(OH)_2_D regulates gene expression by either activating or suppressing specific genes [51,52]. Moreover, 1,25(OH)_2_D shows an approximately 1000 times greater binding affinity to VDR compared to 25(OH)D. The enzyme CYP27B1 is also expressed in various tissues, including activated macrophages, microglia, parathyroid glands, breast tissue, colon, and keratinocytes, where 1,25(OH)_2_D exerts its effects both autocrinally and paracrinally [53].

Traditionally, vitamin D is well-known for its critical role in bone health. However, its influence extends far beyond skeletal functions. Vitamin D is also crucial in supporting immune system regulation. Its effects on various bodily systems are facilitated by the presence of VDR and hydroxylation enzymes in tissues such as the pancreas, kidneys, muscles, and liver. Vitamin D supplementation demonstrates a wide array of beneficial properties, including hormonal, anti-inflammatory, anti-apoptotic, anti-fibrotic, antioxidant, and immune-modulatory effects. Furthermore, it plays a significant role in insulin sensitivity by reducing the expression of pro-inflammatory cytokines such as interleukin-1 (IL-1) and IL-6 [54].

Vitamin D also plays a critical role in glucose metabolism, insulin sensitivity, and body weight regulation [55,56]. Its biological actions are primarily mediated through its interaction with the vitamin D nuclear VDR, a transcription factor that regulates the expression of key genes involved in calcium homeostasis and metabolic pathways, including glucose and lipid metabolism [57,58]. The enzyme 1α-hydroxylase CYP27B1 is expressed in T and B cells with specific immune characteristics that facilitate the regulation of vitamin D levels [59,60,61]. Additionally, 1,25(OH)_2_D_3_ at the level of dendritic cells inhibits inflammatory markers leading to a state of anti-inflammatory immune tolerance characterized by an increased production of interleukin-10 (IL-10) and a decrease in interleukin-12 (IL-12) [62].

On the other hand, 1,25(OH)_2_D_3_ participates in the differentiation of macrophages as a key link in the inflammatory process through the VDR-PPARγ signaling pathway, its main role being the facilitation of the transition from the inflammatory phenotype (M1) to the anti-inflammatory phenotype (M2) [63]. These interactions are of great relevance for the pancreatic inflammatory process of type 1 diabetes, where T and B lymphocytes, as well as macrophages, can be modulated by high doses of vitamin D. Likewise, analogous elements can reduce chronic inflammation by generating a reduction in effector T cells and a subsequent increase in regulatory type cells [64]. It is worth noting that some of the described anti-inflammatory effects may also be relevant to type 2 diabetes.

Figure 1 illustrates the metabolic process of vitamin D with its enzymatic and molecular interactions. 

### 3.1. Glucose Metabolism and Insulin Sensitivity

Vitamin D also has various pleiotropic effects which directly impact metabolic pathologies that alter glucose levels (insulin resistance) such as non-alcoholic fatty liver, metabolic syndrome and type 2 diabetes [65,66,67]. In this context, an inverse relationship has been demonstrated between the Homeostatic Model for the Evaluation of Insulin Resistance (HOMA-IR) and plasma vitamin D deficiency as a homeostatic modeler based on body mass index [67,68].

On a molecular level, vitamin D plays a key role in glucose metabolism and insulin sensitivity, acting directly on pancreatic beta cells, responsible for insulin secretion in response to blood glucose levels [69,70,71,72]. The main mechanism involves the binding of vitamin D to its VDR, which induces the expression of various genes involved in insulin signaling pathways, such as those regulating GLUTs, leading to an increase in glucose uptake in skeletal muscle cells and other peripheral tissues, thus improving glucose control [73,74]. VDR activation stimulates the transcription of specific enzymes and proteins essential for glucose metabolism, optimizing glucose uptake and insulin sensitivity in tissues such as muscle, adipose tissue, and liver cells [75].

However, it is important to note that some individuals do not adequately respond to vitamin D supplementation due to specific genetic variations, such as single nucleotide polymorphisms (SNPs), which may limit the supplement’s efficacy [76]. These genetic markers enable the optimization of nutritional programming by personalizing vitamin D supplementation in patients with type 2 diabetes. While genetic predisposition may influence individual responses to vitamin D supplementation, current evidence suggests that its effects on prediabetes prevention and management are broad and not necessarily restricted to specific genetic groups. Although polymorphisms such as rs10877012 G/T in CYP27B1 have been associated with variations in serum 25-hydroxyvitamin D levels, genetic testing is not yet a routine clinical practice and may add complexity and cost to intervention strategies. Given that vitamin D supplementation has shown general benefits in some areas of diabetes prevention and management, further research is needed to determine the extent to which genetic factors should guide personalized treatment approaches [77].

Vitamin D deficiency also interferes with glucose tolerance, leading to impaired insulin action. This condition is associated with insulin resistance at the level of muscle, adipose and liver cells, making it difficult to eliminate blood glucose [78,79,80].

Regarding insulin receptor expression, vitamin D has been found to increase insulin receptor expression in muscles, liver, and adipose tissue, thereby improving insulin sensitivity [68]. Vitamin D enhances the expression of the insulin receptor in adipose tissue and liver muscle cells by acting as an epigenetic regulator. It promotes the activity of insulin receptor substrates, resulting in a documented increase of more than twofold [68]. In addition, vitamin D also improves insulin receptor sensitivity by activating glucose transport mechanisms, as well as enhancing the conversion of proinsulin to insulin [81,82,83].

Vitamin D deficiency increases the expression of proinflammatory cytokines, which could be the cause of insulin resistance in patients with relatively higher BMI. Obesity is associated with vitamin D hypovitaminosis due to three main reasons: lower sun exposure, low vitamin D intake through nutrition, and sequestration of vitamin D in adipose tissue [74]. In addition to these factors, the regulation of metabolic functions can be further disrupted by adiposity, which affects the secretion of key hormones, including leptin. Leptin, which is secreted by adipose tissue, plays a critical role in appetite regulation and energy balance. Its levels are typically elevated in obese individuals, and this deregulation is linked to insulin resistance. Interestingly, some research suggests that high doses of vitamin D could reduce leptin levels, potentially aiding in the management of body weight and improving insulin sensitivity in patients with insulin resistance. This effect may be mediated through vitamin D binding to its receptors in the paraventricular nucleus of the hypothalamus, influencing caloric intake and weight control mechanisms [74]

Vitamin D also exerts anti-inflammatory effects by modulating the production of proinflammatory cytokines, such as IL-6 and tumor necrosis factor-alpha (TNF-α), which play a significant role in the development of metabolic dysfunction [12,13]. Additionally, vitamin D influences insulin sensitivity through its ability to regulate inflammation and metabolic processes within adipose tissue. It improves the secretion of adipokines like leptin and adiponectin, crucial hormones in weight regulation and metabolic balance [58,84]. Adiponectin has anti-inflammatory properties and promotes insulin sensitivity, and its levels are positively correlated with adequate vitamin D status [85].

Vitamin D plays a significant role in glucose metabolism and insulin sensitivity at a molecular level through its interaction with key metabolic regulators, including AMP-activated protein kinase (AMPK) and mammalian target of rapamycin (mTOR). AMPK is a key energy sensor enzyme that responds to low cellular energy levels, promoting catabolic processes such as fatty acid oxidation and glucose uptake to restore cellular energy balance [86,87]. Once activated, AMPK improves insulin sensitivity by facilitating glucose uptake in muscle and liver cells, thus contributing to better glucose control. Recent studies suggest that vitamin D, through its binding to VDR, can activate AMPK, particularly under metabolic stress conditions. This activation leads to increased fatty acid oxidation and a reduction in lipid accumulation, mitigating insulin resistance risk [88,89].

On the other hand, mTOR is a central controller of cellular growth and metabolism, integrating nutritional signals, energy status, and growth factors. mTORC1 promotes anabolic processes, such as protein synthesis and lipid storage, under nutrient-rich conditions, while its inhibition helps conserve energy during metabolic stress. Dysregulation of mTOR signaling is implicated in metabolic disorders, such as obesity and insulin resistance. It has been shown that vitamin D modulates mTOR activity, potentially reducing inflammation and improving insulin sensitivity by preventing excessive mTORC1 activation in adipose tissue. By influencing mTOR signaling, vitamin D may help counteract excessive fat accumulation and mitigate insulin resistance, playing a protective role in metabolic health [90,91,92,93].

Additionally, vitamin D interacts with mTORC2 and the PI3K/Akt pathway, which play key roles in insulin sensitivity. mTORC2 directly regulates the translocation of GLUT4 to the cell membrane in muscle and adipose tissue, facilitating glucose uptake [94]. Through VDR, vitamin D can modulate mTORC2 activity, promoting glucose homeostasis. Furthermore, the PI3K/Akt pathway, essential for insulin signaling, is activated by vitamin D, improving Akt phosphorylation and optimizing insulin responses in hepatic and muscular cells. Recent studies indicate that these effects may be further enhanced in individuals with favorable polymorphisms in PI3K/Akt pathway genes, suggesting opportunities for personalized interventions [95,96].

The interaction between AMPK and mTOR is critical for regulating energy balance and glucose metabolism. The ability of vitamin D to modulate these pathways provides a molecular mechanism through which it can improve insulin sensitivity and mitigate metabolic dysfunction. Understanding these interactions at the molecular level deepens our knowledge of the role of vitamin D in metabolic health and highlights its potential as a therapeutic agent for preventing and managing insulin resistance and related metabolic disorders.

In the context of glucose metabolism and insulin sensitivity, vitamin D not only interacts with VDR but also regulates specific metabolic pathways, acting as an epigenetic modulator influencing microRNA (miRNA) expression [97]. miRNAs are small non-coding RNAs that play a crucial role in post-transcriptional regulation of inflammation, insulin sensitivity, and adipogenesis, processes fundamental in the pathophysiology of type 2 diabetes [98].

Vitamin D has been shown to regulate the expression of key miRNAs, such as miR-21 and miR-155, both of which are associated with inflammation and insulin signaling. miR-21, when suppressed by vitamin D, decreases nuclear factor κB (NF-κB) activation, a protein essential in inflammatory signal transduction, thus reducing systemic inflammation [99,100]. This epigenetic regulation occurs through the binding of VDR to promoter elements of NF-κB-related genes, inhibiting their expression and subsequent inflammatory cascade. Additionally, miR-155, known to promote a pro-inflammatory phenotype in macrophages, is also negatively regulated by vitamin D, favoring polarization toward an anti-inflammatory (M2) profile, crucial for immune response control in adipose tissue [101,102].

These epigenetic interactions highlight how vitamin D acts as a key modulator at the cellular level, directly regulating critical pathways in the pathophysiology of type 2 diabetes. However, further exploration is needed in future studies to understand how vitamin D supplementation can modulate this complex miRNA network in humans and its long-term clinical impact. Specific studies are required to elucidate the exact interaction pathways between VDR, miRNAs, and other epigenetic factors, as well as their implications for glucose metabolism and insulin sensitivity in diabetic populations.

Figure 2 illustrates the intricate role of vitamin D in glucose metabolism and insulin sensitivity at the molecular level. It highlights how vitamin D interacts with key metabolic pathways, including the AMPK and mTOR pathways.

### 3.2. Vitamin D and HbA1c

Evidence has shown that the inclusion of foods rich in vitamin D can modulate HbA1c levels in diabetic patients, showing a significant relationship between vitamin D intake and glycemic regulation. For instance, the consumption of dairy products has been associated with a reduction of approximately 1% in HbA1c levels, particularly when these products are enriched with both vitamin D and probiotics [104].

This effect is observed particularly through its interaction with metabolic factors such as BMI and place of residence, which also play a crucial role in glucose homeostasis [105]. However, the effect of vitamin D supplementation remains a topic of debate due to the heterogeneity in doses used and intervention periods, which vary considerably between studies. This inconsistency could be explained by the presence of cofactors specific to type 2 diabetes, such as impaired fasting glucose or altered glucose, which are not always accompanied by clinical obesity, complicating the extrapolation of precise results [106].

Furthermore, maintaining healthy lifestyle habits, including a balanced diet and physical activity, is essential to achieve adequate control of type 2 diabetes and improve overall health, particularly in relation to vitamin D levels [107]. In this sense, stress also emerges as a relevant factor, given that the expression of the active form of vitamin D (1,25(OH)2D3) and the enzyme 1-alpha-hydroxylase are linked to stress metabolism, which could have pleiotropic influence on the modulation of type 2 diabetes through stress [108].

Additionally, recent data suggest that the relationship between urinary albumin excretion and vitamin D levels is independent of glucose control, indicating that vitamin D status could have a direct impact on renal function and glucose homeostasis, beyond its influence on blood glucose regulation [109]. Therefore, the inclusion of vitamin D-rich supplements or foods in patients with type 2 diabetes, while promising, should be understood as a complement to a proper diet and exercise plan, not as a substitute [104].

Oxidative stress and mitochondrial dysfunction are key pathophysiological features in the progression of type 2 diabetes, playing significant roles in the development of insulin resistance and cellular damage. Vitamin D, through activation of its VDR, regulates the expression of crucial endogenous antioxidants such as superoxide dismutase (SOD) and catalase, which are enzymes responsible for neutralizing reactive oxygen species (ROS). By reducing ROS levels, vitamin D helps mitigate oxidative stress, preventing damage to cellular structures, including lipids, proteins, and DNA. Moreover, vitamin D plays a critical role in promoting mitochondrial biogenesis—the process by which new mitochondria are formed—through the activation of peroxisome proliferator-activated receptor gamma coactivator 1-alpha (PGC-1α). PGC-1α is a key coactivator involved in mitochondrial energy production, and its activation by vitamin D enhances the efficiency of the mitochondrial network, which is crucial for maintaining energy balance in cells [110,111].

Vitamin D deficiency has been linked to mitochondrial dysfunction, characterized by increased mitochondrial fragmentation, reduced mitochondrial size, and impaired activity of the respiratory chain complexes. This dysfunction results in decreased ATP production, the primary energy currency of cells, which directly contributes to insulin resistance. The reduced mitochondrial function in response to vitamin D deficiency exacerbates the development of insulin resistance, particularly in tissues that are highly metabolically active, such as skeletal muscle and the liver [109,112]. In these tissues, mitochondria play a critical role in maintaining glucose metabolism and energy production. A decline in mitochondrial efficiency leads to increased oxidative stress, which not only impairs insulin signaling pathways but also promotes inflammatory responses, further contributing to the progression of type 2 diabetes.

These findings highlight the dual role of vitamin D as both an antioxidant and a pro-mitochondrial agent. Its ability to modulate oxidative stress and enhance mitochondrial function presents a promising therapeutic strategy to restore metabolic balance in individuals with type 2 diabetes. Targeting these mechanisms through vitamin D supplementation could potentially improve insulin sensitivity, reduce cellular damage, and enhance mitochondrial health, offering a valuable approach for the management and prevention of type 2 diabetes [113,114]. Future studies are essential to elucidate the precise molecular pathways by which vitamin D interacts with mitochondria and oxidative stress networks, as well as to determine its long-term clinical impact on metabolic health in diabetic populations. Additionally, nutritional interventions should consider these findings to personalize the strategy for managing type 2 diabetes, addressing not only vitamin D intake but also other metabolic and behavioral factors associated with the disease.

### 3.3. Adipogenesis

Vitamin D plays a critical role in the regulation of fat storage and energy homeostasis, contributing significantly to key metabolic processes. It activates the peroxisome proliferator-activated receptor gamma (PPAR-γ), a crucial regulator in adipocyte differentiation and lipid metabolism. This activation promotes the redistribution of fat toward more metabolically active deposits [115,116,117]. This receptor controls both the formation of adipose tissue and lipid storage, favoring the development of metabolically active adipocytes and preventing excessive fat accumulation [118,119,120,121].

Adipogenesis is the differentiation process by which mesenchymal stem cells (MSCs) develop into functional adipocytes, characterized by specific phenotypic features. During this process, MSCs respond to extracellular signals and undergo stages of proliferation and clonal expansion, producing preadipocytes—cells with high plasticity that later differentiate into mature adipocytes [122,123]. Adipocytes are spherical in appearance, ranging from 10 to 100 µm in diameter, containing clustered organelles and a nucleus that is displaced towards the cell periphery due to a unilocular triglyceride vesicle occupying most of the cytoplasm, thus limiting the presence of other organelles such as mitochondria and the Golgi apparatus [124].

Adipogenesis occurs in several well-defined stages, starting with the conversion of MSCs into preadipocytes. Although these preadipocytes are not morphologically distinct from their progenitor cells, they exhibit the activation of specific transcription factors. In the first phase, AP-1 family factors and C/EBPβ and C/EBPδ are activated. These proteins induce a crucial second step, activating key differentiation genes such as PPARγ and C/EBPα, which together regulate adipocyte maturation and function [125].

Adipogenesis plays a significant role in maintaining metabolic health, particularly in processes associated with obesity. Understanding the mechanisms and regulators of adipogenesis is essential for developing effective strategies to improve metabolic health. Adipogenesis can counteract the harmful metabolic effects of overweight and obesity by generating new adipocytes rather than expanding existing ones, which tend to have a proinflammatory and hypoxic profile. The expansion of adipose tissue through adipogenesis results in a healthier profile, characterized by smaller and more numerous adipocytes, with reduced inflammation and fibrosis, contributing to better metabolic health [126,127,128,129,130].

In clinical studies, meta-analyses have shown a modest effect of vitamin D supplementation on body measurements such as weight, waist circumference, and hip circumference. These findings are important as they link basic molecular mechanisms, such as the regulation of adipocyte differentiation and function, to tangible clinical metrics. For example, some studies indicate that vitamin D supplementation can reduce body weight by around 2–3% in individuals with low baseline vitamin D levels, suggesting a potential role in improving adipose tissue distribution [131,132,133].

Multiple factors and events regulate adipogenesis and may contribute to the etiology of obesity [134]. The differentiation and transformation process of mesenchymal stem cells, driven by the cascade of events triggered by regulatory keys, produces three main types of mature adipocytes. These are white adipocytes, the primary function of which is the storage and reserve of triglycerides for energy use; brown adipocytes, which are responsible for heat production through the process of thermogenesis; and beige adipocytes, which are commonly considered an intermediate form between white and brown adipocytes, with the capacity for both storage and heat production [127].

Clinical evidence has suggested that vitamin D may promote the differentiation of beige adipocytes, which have both storage and thermogenic properties, providing insight into its potential role in body fat regulation [131,132,133]. Furthermore, vitamin D plays a key role in autophagy, a crucial cellular mechanism for energy balance and cellular homeostasis. Through the activation of the VDR, vitamin D regulates the expression of genes involved in adipocyte differentiation and the activation of autophagy processes in adipose tissue. Excessive adipogenesis can lead to the formation of poorly functional adipose tissue, triggering chronic inflammation. By influencing these processes, vitamin D helps maintain a healthy balance in the formation of adipocytes, preventing the accumulation of unhealthy fat and reducing inflammation associated with obesity [135,136].

To gain a deeper understanding of the process of adipogenesis, Figure 3 visually illustrates the key stages of adipocyte differentiation, showing how mesenchymal stem cells (MSCs) transform into mature adipocytes through the activation of essential regulatory factors.

Complementarily, Table 2 provides a detailed breakdown of the key genetic regulators involved in this process, including their specific functions and associated characteristics. Together, these tools offer an integrated approach to analyzing the molecular mechanisms underlying adipocyte differentiation, such as the activation of PPARγ, C/EBPα, and other critical factors that guide the formation of different types of mature adipocytes: white, brown, and beige. Both visual aids are essential for understanding how adipocyte formation is regulated, influencing lipid storage, heat production, and energy metabolism, as well as its impact on metabolic health.

### 3.4. Vitamin D Levels

Serum levels of vitamin D can be reported in ng/mL or nmol/L using the coefficient of variation of the 25(OH)D test, although there is no single criterion to establish the cutoff points that determine these serum states [146]. Commonly accepted values in ng/mL or nmol/L are those reported by the Institute of Medicine of the United States, which define severe deficiency (10–12 ng/mL or 25–30 nmol/L), slight deficiency (<20 ng/mL or <50 nmol/L), and adequate status (>20 ng/mL or >50 nmol/L) [2].

In this sense, the lack of a single criterion has led to the absence of consensus regarding vitamin D levels, which has generated discrepancies between the various recommendations provided by institutions. Although the Institute of Medicine of the U.S. and other organizations, such as the International Osteoporosis Foundation and the American Geriatrics Society, define different levels, these variations are due to the interpretation of the available evidence, whose results are limited by multiple factors, such as methodological differences in measurement protocols, the populations studied, and the bone health criteria, as well as genetic, environmental, and cultural variability [146]. These discrepancies also reflect that certain levels may be more appropriate for specific patient populations [2,3,34,38,39,146]. For example, higher vitamin D levels might be recommended for older adults or those with certain chronic conditions, while lower levels may be considered adequate for healthy individuals without deficiencies.

Vitamin D deficiency is a widely recognized public health issue with significant prevalence across various populations. It is important to highlight those individuals living in northern latitudes, where sun exposure is limited, are more susceptible to this deficiency due to reduced cutaneous synthesis of vitamin D. Additionally, individuals with darker skin, who have higher concentrations of melanin, have a reduced ability to synthesize vitamin D in response to ultraviolet radiation from the sun [147,148,149]. These combined factors contribute to a higher prevalence of vitamin D deficiency in these groups, which can have important implications for metabolic health and the risk of chronic diseases such as type 2 diabetes. The lack of adequate sun exposure and the genetic characteristics of the skin can be key factors that modulate vitamin D levels in the population, underscoring the need for prevention and treatment strategies in these areas.

Table 3 presents an overview of serum vitamin D levels, highlighting the commonly used cutoff points to categorize vitamin D status. It discusses how vitamin D levels can be measured in ng/mL or nmol/L using the 25(OH)D test, and outlines accepted thresholds for deficiency, sufficiency, and adequacy.

The relationship between vitamin D, AMPK, mTOR, and obesity involves intricate molecular interactions that regulate energy metabolism and adiposity. Vitamin D plays a crucial role in these processes, influencing both AMPK and mTOR signaling pathways. Deficiency in vitamin D has been linked to increased visceral adiposity, a major risk factor for metabolic diseases. Excessive fat accumulation in the abdominal region is closely tied to insulin resistance and metabolic dysfunction, which are further exacerbated by the chronic inflammatory state induced by vitamin D deficiency [86,87,88,89,90,91,92,93].

Vitamin D deficiency impairs the activation of AMPK, an important energy sensor that promotes fatty acid oxidation and prevents lipid accumulation. Through VDR-mediated signaling, vitamin D can enhance AMPK activity, leading to improved energy balance and reduced fat storage [89]. Simultaneously, vitamin D inhibits mTOR activity, which plays a key role in lipid synthesis and cell growth. Elevated mTOR activity contributes to increased fat deposition, insulin resistance, and the development of obesity. Vitamin D’s ability to suppress mTOR activity helps prevent excessive fat accumulation and lowers the risk of obesity and its associated metabolic disorders [150].

Recent studies also highlight the role of vitamin D in modulating inflammation in adipose tissue. Through its interaction with mTOR, vitamin D reduces chronic inflammation, which is a key driver of obesity-related metabolic diseases. By decreasing inflammation, vitamin D contributes to healthier adipose tissue function and lowers the risk of metabolic syndrome and obesity [150].

However, excessive vitamin D intake, while rare, can lead to toxicity and hypercalcemia, commonly resulting from excessive dietary supplementation. Overactivation of parathyroid hormones due to high vitamin D levels can cause disturbances in cardiac rhythm, confusion, and disorientation [151,152]. Therefore, maintaining optimal levels of vitamin D is essential to balance its metabolic benefits while avoiding adverse effects.

To prevent vitamin D deficiency, various practical interventions can be implemented. Supplementation is a key strategy for individuals at risk, such as those with limited sun exposure or clinically diagnosed deficiency. Regular supplementation with vitamin D contributes significantly to improving vitamin D levels in vulnerable populations, reducing the risk of metabolic diseases associated with it [153,154,155,156]. Similarly, dietary changes play a crucial role, as consuming foods rich in vitamin D, such as fatty fish, eggs, liver, and fortified dairy products, can significantly increase daily intake and help reduce the prevalence of hypovitaminosis [156].

Controlled sun exposure is another effective preventive intervention. The human body synthesizes vitamin D when exposed to sunlight. However, depending on multiple factors such as the time of day, season, skin pigmentation, cloud cover, and use of sunscreen, this process can be modulated. It also requires precautions to avoid acute cases of sunburn or solar burns, as well as long-term damage related to melanoma [157,158].

From a public health perspective, the most comprehensive strategies involve government policies for the fortification of foods, such as vitamin D-fortified milk, which has resulted in a significant reduction in deficiency rates in the general population [159,160,161]. This approach has proven effective in reducing related metabolic diseases, such as osteoporosis [162]. Therefore, the prevention of vitamin D deficiency requires a multifaceted approach that combines supplementation, dietary changes, controlled sun exposure, and public health policies to improve metabolic health and reduce the risk of chronic diseases.

### 3.5. Recommendations for Vitamin D Intake and Considerations for Vulnerable Populations

Vitamin D plays a vital role in bone health, immune function, and overall metabolism. Adequate intake is essential for the general population, as well as for individuals who are more vulnerable due to physiological, geographical, or socioeconomic factors. Below are tailored recommendations and preventive strategies for various population groups.

The recommended intake of vitamin D varies according to age, health status, and deficiency risk. For infants, a daily intake of 10 μg is advised during the first year of life. In children and adolescents, requirements gradually increase, stabilizing at 15 μg per day from ages 9 to 70. For individuals over 70 years, the recommended intake rises to 20 μg to compensate for reduced cutaneous synthesis and to prevent conditions such as osteoporosis, sarcopenia, and functional decline [2,163].

At-risk populations, including those with chronic diseases or conditions that impact vitamin D metabolism (e.g., malabsorption, renal insufficiency), are advised to supplement 1000 to 2000 IU daily, depending on baseline serum levels. In specific clinical scenarios, these doses may be increased up to 10,000 IU under medical supervision. It is crucial to monitor potential drug interactions, particularly with corticosteroids, as these can exacerbate deficiency [2,163].

In regions with limited sun exposure, such as Nordic countries, vitamin D deficiency is more prevalent during the winter months. Policies promoting the fortification of staple foods, such as milk and oils, combined with seasonal supplementation, have proven effective in preventing hypovitaminosis D. In these areas, ensuring adequate intake during periods of reduced UVB radiation is especially critical to avoid complications related to deficiency [2,163].

Regarding vitamin D toxicity, it remains a significant concern due to the increasing availability and consumption of high-dose supplements. It is essential to emphasize the safe upper limit for daily vitamin D intake. According to current evidence, the safe daily upper intake level of vitamin D for most adults is 4000 IU [163]. Exceeding this threshold, particularly over prolonged periods, can pose risks and increase the likelihood of vitamin D toxicity. Regular monitoring of serum vitamin D and calcium levels, particularly in individuals consuming high doses, is essential to mitigate these risks. Additionally, dose adjustments based on individual needs and current health status are crucial to ensure safe supplementation and prevent toxicity. Doses exceeding 300,000 IU bolus have been associated with a higher risk of hypercalcemia and hypercalciuria, and in general, should be avoided as a standard practice [164].

Table 4 provides a comprehensive overview of the recommended intake of vitamin D, highlighting tailored strategies for different population groups, including those at higher risk due to physiological, geographical, or socioeconomic factors. It outlines daily intake guidelines based on age and health status, emphasizing the increased needs for older adults and populations with conditions that impact vitamin D metabolism, such as chronic diseases or malabsorption issues. The recommendations presented are primarily based on those from the Institute of Medicine, which provides general guidelines for the broader population [2]. However, alternative guidelines from organizations like the International Osteoporosis Foundation and the American Geriatrics Society suggest higher thresholds for older adults and those with osteoporosis, as they are at greater risk of vitamin D deficiency due to aging and bone health concerns. Additionally, the Endocrine Society recommends higher doses for individuals with metabolic disorders, including diabetes or obesity, where vitamin D plays a crucial role in regulating insulin sensitivity and metabolic health [2,34,38,39,146]. These differing perspectives highlight the importance of considering individual health conditions, life stages, and geographic factors when determining vitamin D needs.

## 4. Limitations of the Current Evidence

The study of the relationship between vitamin D and type 2 diabetes mellitus has significantly progressed in recent decades. However, several limitations hinder a comprehensive understanding of this relationship and its application in public health policies and clinical practice. A critical evaluation of these limitations is essential to identify areas for improvement and guide future research priorities.

One primary challenge is the heterogeneity in study designs, which includes variations in the definitions of prediabetes and type 2 diabetes, study populations, dosages, and durations of vitamin D supplementation. Such variability complicates result synthesis and limits the ability to draw definitive conclusions [5,165,166,167].

Vitamin D exerts significant epigenetic effects through the modulation of DNA methylation, histone acetylation, and the regulation of transcription factors associated with its nuclear receptor, VDR. These epigenetic modifications directly influence genes related to insulin sensitivity and chronic inflammation, both key factors in the progression of type 2 diabetes. Vitamin D can reduce the methylation of the promoter region of the IRS1 gene, improving insulin signaling in peripheral tissues [97,98,99,100,101,102]. Additionally, histone acetylation mediated by the VDR/HDAC interaction affects genes involved in adipocyte differentiation and immune response. Recent studies have identified genetic variants in CYP27B1, responsible for the activation of vitamin D, which interact with these epigenetic modifications, modulating the clinical response to supplementation [49,53,59,60].

Vitamin D modulates microbial composition by promoting the growth of beneficial bacteria such as Bifidobacterium and Lactobacillus, while reducing the proportion of pathogenic species. It also regulates intestinal barrier integrity by increasing the expression of tight junction proteins, such as claudins and occludins, preventing the translocation of bacterial lipopolysaccharides that trigger chronic inflammation [33]. The relationship between vitamin D and the intestinal microbiota has emerged as a critical axis in the regulation of systemic inflammation and insulin sensitivity in type 2 diabetes. Recent studies suggest that these effects of vitamin D are mediated by microbial metabolites such as short-chain fatty acids, which improve insulin sensitivity and local inflammation in adipose tissue [168].

On the other hand, mTOR is a central controller of cellular growth and metabolism, integrating nutritional signals, energy status, and growth factors. mTORC1 promotes anabolic processes, such as protein synthesis and lipid storage, under nutrient-rich conditions, while its inhibition helps conserve energy during metabolic stress. Dysregulation of mTOR signaling is implicated in metabolic disorders, such as obesity and insulin resistance. It has been shown that vitamin D modulates mTOR activity, potentially reducing inflammation and improving insulin sensitivity by preventing excessive mTORC1 activation in adipose tissue [90,91,92,93].

The methodologies used to measure serum vitamin D levels (25[OH]D) lack standardization. Techniques such as high-performance liquid chromatography (HPLC) and immunoassays often produce varying results due to differences in sensitivity and precision. This inconsistency not only impedes the comparability of findings across studies but also poses challenges in establishing universally accepted thresholds for vitamin D sufficiency, deficiency, and toxicity [146,169].

Uncontrolled confounding variables, such as physical activity, dietary patterns, BMI, sun exposure, and genetic predisposition, further complicate interpretation of results. These factors influence both vitamin D levels and the risk of type 2 diabetes, making it difficult to isolate the independent effects of vitamin D supplementation [170,171,172,173]. Moreover, vitamin D metabolism and its effects on glucose homeostasis can be influenced by genetic, environmental, and cultural factors, although the extent and nature of these interactions remain uncertain [174,175]. Most studies focus on short- to medium-term outcomes of vitamin D supplementation, leaving a gap in understanding its long-term impact on type 2 diabetes prevention and management. Differences in the form, dosage, and frequency of vitamin D supplementation across studies introduce additional variability. Studies employing vitamin D3 may yield different outcomes compared to those using vitamin D2 [2,151].

The interactive effects of vitamin D supplementation with other lifestyle or pharmacological interventions, such as exercise or metformin use, remain insufficiently studied. Investigating these interactions could provide a more holistic understanding of vitamin D’s contributions to metabolic health [165]. In this context, the ADA does not recommend routine supplementation or a specific dose of vitamin D for the management of type 2 diabetes unless a deficiency is diagnosed. Its recommendations are limited to indicating a potential association between low vitamin D levels and an increased risk of diabetes, but it considers that the evidence is not yet sufficient to justify its use as a preventive or therapeutic measure. Therefore, its guidelines only apply to individuals with vitamin D deficiency or those at risk, such as older adults, pregnant women, or individuals on restrictive diets, whose needs may require supplementation [3,33,37,38].

Addressing these limitations will enable future research to offer clearer guidance on the role of vitamin D in preventing and managing type 2 diabetes, enhancing its relevance and applicability in clinical and public health contexts.

Table 5 provides a comprehensive summary of the current evidence exploring the impact of vitamin D on type 2 diabetes. This compilation includes various studies assessing the efficacy of vitamin D supplementation in reducing diabetes risk, improving glycemic control, and examining its association with long-term health outcomes.

## 5. Clinical Implications and Future Research Directions

Research on the relationship between vitamin D and type 2 diabetes faces significant challenges that limit the extrapolation of findings and their clinical applicability. A primary obstacle lies in the genetic variability across studied populations, as polymorphisms in genes related to the VDR and enzymes involved in its metabolism substantially influence responses to supplementation, complicating the generalizability of results. Furthermore, inconsistencies in methods for measuring vitamin D levels, particularly 25(OH)D, contribute to discrepancies in outcomes. Variations in analytical techniques, such as immunoassays versus mass spectrometry, hinder cross-study comparisons and the interpretation of conclusions. The lack of longitudinal studies evaluating the sustained effects of supplementation and its interaction with lifestyle factors or concomitant treatments further limits a comprehensive understanding of its impact.

Another critical challenge lies in the inadequate adjustment for confounding factors. Variables such as body mass index, physical activity, comorbidities, and medication influence both vitamin D levels and metabolic outcomes, complicating the establishment of clear causal relationships. Addressing these gaps requires more robust methodological approaches and advanced technologies to generate reliable and generalizable data. Integrating personalized medicine models, such as genotyping to identify genetic variants associated with vitamin D metabolism, is essential for stratifying participants and tailoring recommendations. Additionally, detailed analyses by ethnicity, sex, and sociodemographic factors are critical for personalizing intervention strategies.

The standardization of measurements and protocols must be prioritized. This includes adopting universally accepted methods, such as mass spectrometry, for assessing 25(OH)D levels and developing international guidelines that consistently define thresholds for vitamin D deficiency, insufficiency, and sufficiency. Clinical trials should adhere to homogeneous protocols regarding dosages, forms of supplementation, and follow-up durations. Advanced technologies like CRISPR-Cas9 offer transformative potential to explore the role of VDR and its genetic variants in in vitro models, while proteomics and metabolomics can help identify precise biomarkers for the impact of vitamin D on glucose metabolism and inflammation. Artificial intelligence may also play a pivotal role by analyzing large datasets and modeling complex interactions among genetic, environmental, and metabolic factors.

The design of longitudinal and controlled studies with diverse population samples is necessary to evaluate the sustained effects of vitamin D on type 2 diabetes management. Such studies should account for interdependent variables, including the combined impact of vitamin D supplementation and physical exercise, to provide more comprehensive conclusions. In vulnerable populations, such as those with limited sunlight exposure or access to fortified foods, culturally relevant and sustainable supplementation strategies are essential, considering factors such as body composition, micronutrient interactions, and potential adverse effects. Evaluating combinations of vitamin D with nutrients like calcium or magnesium could optimize metabolic benefits and personalized medical interventions; utilizing tools such as artificial intelligence, metabolomics, and proteomics could have a lasting impact on public health and should therefore be a priority.

These approaches aim to overcome current limitations and advance our understanding of the role of vitamin D in type 2 diabetes prevention and management, facilitating the implementation of strategies based on robust evidence.

## 6. Conclusions

This review consolidates evidence on vitamin D’s multifaceted role in the prevention and management of type 2 diabetes, highlighting its molecular mechanisms and clinical implications. By modulating inflammation, enhancing glucose metabolism, and supporting healthier adipocyte profiles, vitamin D emerges as a key regulator of metabolic health. However, the variability in study designs, inconsistent methodologies for measuring serum levels, and population-specific responses underscore the challenges in translating these findings into universal clinical guidelines. To optimize vitamin D’s therapeutic potential, future research must prioritize standardized methodologies, long-term trials, and individualized approaches that consider genetic, environmental, and socioeconomic factors. Additionally, integrating vitamin D supplementation with lifestyle modifications and pharmacological interventions may offer synergistic benefits, paving the way for comprehensive strategies to mitigate the burden of type 2 diabetes. This review reaffirms the importance of vitamin D as an adjunct in type 2 diabetes management and advocates for targeted research to bridge gaps and maximize its translational impact.

## Figures and Tables

**Figure 1 ijms-26-02153-f001:**
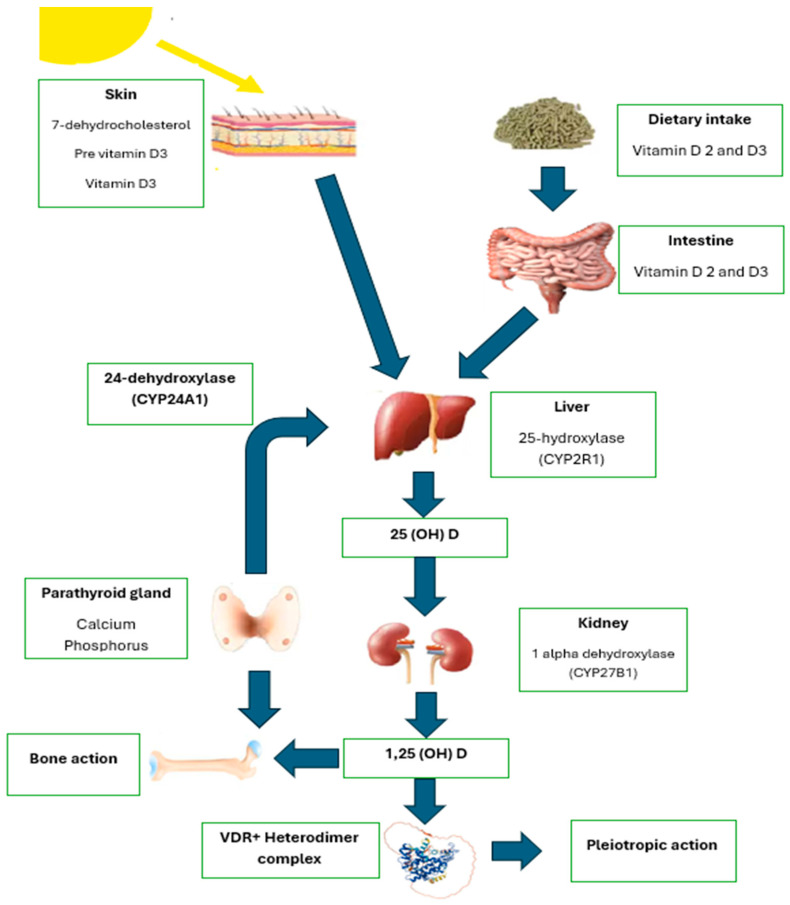
Vitamin metabolism. Source: adaptation of Fuentes-Barría et al. [49].

**Figure 2 ijms-26-02153-f002:**
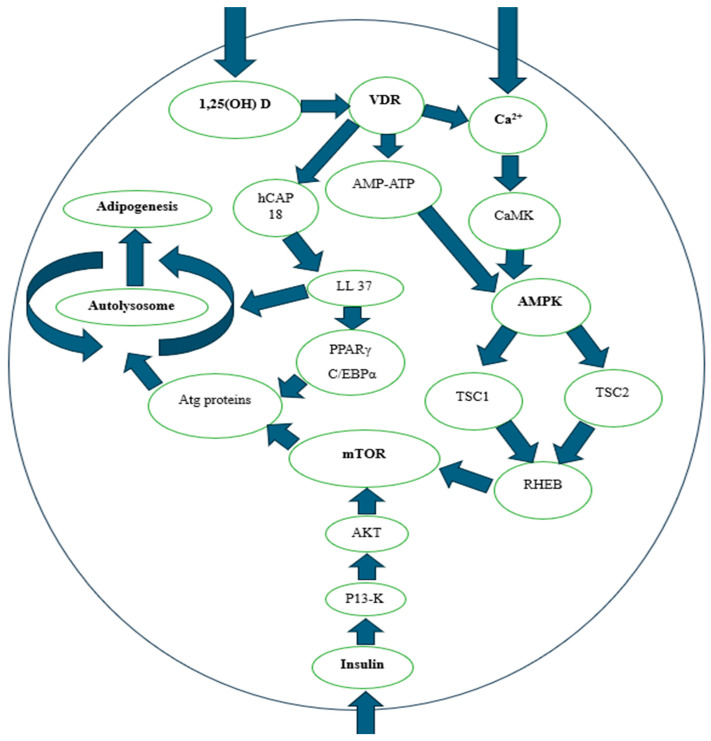
Regulation of AMPK and mTOR. 1,25 OH (D): 1,25-dihydroxyvitamin D, VDR: Vitamin D Receptor, Ca^2+^: Calcium, hCAP18: Human Cathelicidin Antimicrobial Peptide 18, AMP-ATP: Adenosine Monophosphate and Adenosine Triphosphate Relationship, CaMK: Calcium/Calmodulin-Dependent Protein Kinase, LL37: Leucina-Leucina 37, AMPK: AMP-activated protein kinase, TSC1: Tuberous Sclerosis Complex 1, TSC2: Tuberous Sclerosis Complex 2, RHEB: Ras Homolog Enriched in Brain, mTOR: Mechanistic target of rapamycin, PPARγ: Peroxisome Proliferator-Activated Receptor Gamma, C/EBPα: CCAAT/Enhancer Binding Protein Alpha, P13-K: Phosphoinositide 3-Kinase, AKT: Protein Kinase B. Source: adaptation Wan et al. [103].

**Figure 3 ijms-26-02153-f003:**
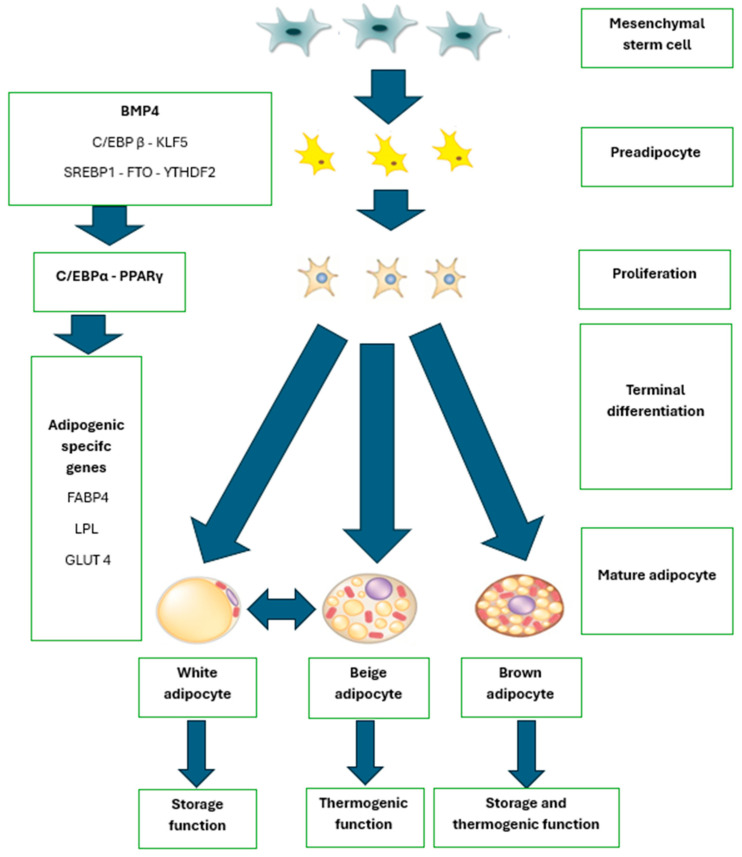
Adipogenesis. BMP4: Bone Morphogenetic Protein 4, C/EBP β: CCAAT/Enhancer Binding Protein Beta, KLF5: Kruppel-Like Factor 5, SSREBP1: Sterol Regulatory Element-Binding Protein 1, FTO: Fat mass and obesity-associated gene, YTHDF2: YTH N6-Methyladenosine RNA Binding Protein 2, EBPα: CCAAT/Enhancer Binding Protein Alpha, PPARγ: Peroxisome Proliferator-Activated Receptor Gamma, FABP4: Fatty Acid-Binding Protein 4. Source: adaptation of Hafidi et al. [127].

**Table 1 ijms-26-02153-t001:** Diagnostic criteria and risk factors for type 2 diabetes mellitus.

Category	Variable/Factor	Criteria/Description	Ref.
Diagnostic Criteria	Plasma glucose	≥126 mg/dL.	[35]
	Oral glucose tolerance	2 h post-load ≥200 mg/dL.	[35]
	Hemoglobin A1c	≥6.5%.	[35]
	Random plasma glucose	≥200 mg/dL with classic symptoms.	[35]
	C-peptide	Relatively preserved.	[35]
	Ketoacidosis	Less frequent.	[35]
Risk Factors	Obesity	Excess body fat.	[40]
	Sedentary lifestyle	Low physical activity.	[41]
	Family history	Relatives with type 2 diabetes.	[42]
	Age	Predominantly adults (though can present in children).	[43]
	Diet high in fats and sugars	Sugary, fatty diet linked to insulin resistance.	[44]
	Hypertension	High blood pressure.	[45]
	High cholesterol levels	Elevated Low-Density Lipoprotein and Triglycerides.	[46]

**Table 2 ijms-26-02153-t002:** Key regulators of adipogenesis.

Gene	Function	Characteristics	Ref.
*C/EBPα*	Induces PPARγ expression, crucial for early adipocyte differentiation.	Activates genes involved in lipid metabolism.	[113]
*PPARγ*	Regulates fat cell differentiation and adipogenesis.	Activated by lipids, promotes adipocyte differentiation.	[137]
*FABP4*	Modulates adipogenesis by influencing PPARγ activity.	Links metabolism to inflammation, elevated levels associated with obesity and insulin resistance.	[138]
*BMP4*	Regulates precursor cell commitment to adipocytes.	Downregulates PDGFRβ, promotes adipogenic differentiation, induces PPARγ expression.	[139]
*FTO*	Modulates mRNA stability, impacting early adipogenesis.	Demethylase activity affects lipid metabolism and adipocyte differentiation.	[140]
*YTHDF2*	Influences mRNA stability, inhibits adipogenesis.	Degrades m6A-modified mRNAs, affecting cell cycle and differentiation.	[141,142]
*mTOR*	Key regulator of growth and metabolism, affecting adipocyte differentiation.	Coordinates with lysosomes during adipogenesis, regulates energy metabolism and insulin signaling.	[143]
*SSREBP1*	Regulates lipid synthesis during adipogenesis.	Induces PPARγ expression, involved in lipid accumulation and adipogenesis.	[144]
*KLF5*	Induces early adipocyte differentiation, works with other transcription factors.	Part of complex networks influenced by growth factors, circadian proteins, and regulatory molecules.	[145]

PPARγ: Peroxisome Proliferator-Activated Receptor Gamma, C/EBPα: CCAAT/Enhancer Binding Protein Alpha, FABP4: Fatty Acid-Binding Protein 4, BMP4: Bone Morphogenetic Protein 4, FTO: Fat mass and obesity-associated gene, YTHDF2: YTH N6-Methyladenosine RNA Binding Protein 2, mTOR: mechanistic Target of Rapamycin, SSREBP1: Sterol Regulatory Element-Binding Protein 1, KLF5: Kruppel-Like Factor 5.

**Table 3 ijms-26-02153-t003:** Interpretation of serum vitamin D levels.

Levels	Netherlands	Institute of Medicine	International Osteoporosis Foundation and American Geriatrics Society	Expert Opinion
Severe Deficiency	10–12 ng/mL 25–30 nmol/L	10–12 ng/mL 25–30 nmol/L	10–12 ng/mL 25–30 nmol/L	10–12 ng/mL 25–30 nmol/L
Slight Deficiency	N/A	<20 ng/mL <50 nmol/L	<30 ng/mL <75 nmol/L	<40 ng/mL <100 nmol/L
Adequate	>10–12 ng/mL >25–30 nmol/L	>20 ng/mL >50 nmol/L	>30 ng/mL >75 nmol/L	>40 ng/mL >100 nmol/L

N/A: Not applicable. Source: adaptation of Herrera-Molina et al. [146].

**Table 4 ijms-26-02153-t004:** Vitamin D intake recommended.

Years	Institute of Medicine				Deficiency Risk for the Endocrine Society	
	AI (μg/UI)	EAR (μg/UI)	RDA (μg/IU)	UL (μg/IU)	IU	UL (IU)
0 to 0.5	10/400	N/A	N/A	25/1000	400 to 1000	2000
0.5 to 1	10/400	N/A	N/A	38/1500	400 to 1000	2000
1 to 3	N/A	10/400	15/600	63/2500	600 to 1000	4000
4 to 8	N/A	10/400	15/600	75/3000	600 to 1000	4000
9 to 13	N/A	10/400	15/600	100/4000	600 to 1000	4000
14 to 18	N/A	10/400	15/600	100/4000	600 to 1000	4000
19 to 30	N/A	10/400	15/600	100/4000	1500 to 2000	10,000
31 to 50	N/A	10/400	15/600	100/4000	1500 to 2000	10,000
51 to 70	N/A	10/400	15/600	100/4000	1500 to 2000	10,000
>70	N/A	10/400	20/800	100/4000	1500 to 2000	10,000

AI: Adequate Intake, EAR: Estimated Average Requirement, RDA: Recommended Dietary Allowances, UL: Tolerable upper intake level, IU: International Units, μg: microgram, N/A: Not applicable. Source: adaptation of Demay et al. [2].

**Table 5 ijms-26-02153-t005:** Summary of current evidence on type 2 diabetes.

Purpose	Hazard Ratio (95% CI)	Ref
Evaluate whether administration of vitamin D decreases risk for diabetes among people with prediabetes.	0.85 [95% CI, 0.75 to 0.96]	[1]
Assess whether vitamin D supplementation reduces the risk of type 2 diabetes in people with prediabetes.	0.89 (95% CI 0.80 to 0.99; I^2^ = 0%)	[5]
Investigating whether low serum 25OHD can predict the onset of diabetes in prospective studies among older adults.	1.31 (95% CI, 1.11–1.54; I^2^ = 37%)	[8]
Examine the therapeutic effects of vitamin D supplementation versus placebo on glycemic control, pregnancy complications, and newborn outcomes in pregnant women diagnosed with Gestational diabetes mellitus.	−10.20 (95% CI, −13.43 to −6.96, I^2^ = 80%)	[82]
Evaluate the effects of oral vitamin D supplementation on glycemic control in type 2 diabetes patients compared with a placebo, and to assess various factors’ influences on supplementation effects.	−0.57 (95%CI: −1.09 to −0.04; I^2^ = 83%)	[165]
Examining whether hypovitaminosis D can predict incident diabetes in prospective longitudinal studies conducted among older adults.	1.20 (95% CI, 1.06 to 1.35, I^2^ = 29.9%)	[176]
Evaluate the association between vitamin D status and all-cause mortality and cardiovascular disease in people with type 2 diabetes.	1.36 (95% CI, 1.23 to 1.49, I^2^ = 57%)	[177]

## Data Availability

The data related to this study are available in this article.

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
