# Peer review of "Vitamin D and Type 2 Diabetes Mellitus: Molecular Mechanisms and Clinical Implications—A Narrative Review"

_ijms, 2025, doi:10.3390/ijms26052153_

Round 1
Reviewer 1 Report
Comments and Suggestions for Authors
I have read the paper entitled "Vitamin D and Type 2 Diabetes Mellitus: Molecular Mechanisms and Clinical Implications a Narrative Review" by Fuentes-Barría and colleagues. The review states that vitamin D has complementary therapeutic potential in the treatment of type 2 diabetes and identifies critical gaps in research, such as optimal dosing strategies and long-term effects across diverse populations.
The paper is well written, is readeble.
This reviewer has no concerns.
Author Response
The reviewer's positive perception is appreciated.
Reviewer 2 Report
Comments and Suggestions for Authors
This manuscript entitled Vitamin D and Type 2 Diabetes Mellitus: Molecular Mecha- 2
nisms and Clinical Implications a Narrative Review explores the role of vitamin D in managing type 2 diabetes, emphasizing its effects on inflammation, glucose metabolism, and adipocyte function. It is well-written, well-structured, clear, and concise, making it easy to navigate. The figures and tables are well-designed.
Some aspects could be improved:
• The introduction is clear and concise, and the scientific content is well-founded.
• The molecular mechanisms of vitamin D in insulin sensitivity, inflammation, adipose metabolism, and glucose regulation are thoroughly described.
• The discussion on vitamin D’s role in glucose metabolism, oxidative stress, and mitochondrial function is well-articulated.
• Lines 295-297: It is mentioned that vitamin D-rich foods can modulate HbA1c levels, but it would be helpful to specify examples of such foods to provide more clarity.
• Lines 418-424: The variation in values should be clarified, indicating which values are more appropriate for specific patient populations.
• Table 3: Presents the differences between the values recommended by various institutions. However, please clarify why these discrepancies exist. Please add a brief explanation in the text ( what could contribute to these variations?.)
• Table 4: The rationale behind choosing IOM recommendations should be explained. Why IOM and not other guidelines? Or when to choose one over the other? For instance, the Institute of Medicine (IOM) provides generalized reference values for the general population, while the International Osteoporosis Foundation (IOF) and the American Geriatrics Society propose higher thresholds for elderly individuals and those with osteoporosis. Additionally, the Endocrine Society suggests higher doses for individuals with metabolic disorders. Consider adding these perspectives to the text/ or table. Furthermore, the table should specify recommendations based on gender, pregnancy, and lactation (even if the values remain the same). Also, the definition of deficiency risk in the table should be clarified—are these values based also on IOM recommendations?
• Chapter 4: This section is well-structured and highlights the key limitations in drawing definitive conclusions about vitamin D and diabetes. However, I suggest adding a few sentences summarizing the latest position of the American Diabetes Association (ADA) regarding vitamin D supplementation in type 2 diabetes. According to ADA, Is supplementation explicitly recommended for diabetes management / or not? If so, what dose and under what circumstances?
• Chapter 5: This section effectively outlines both the challenges (e.g., lack of long-term studies, confounding factors) and the opportunities (e.g., personalized medicine, artificial intelligence, metabolomics, and proteomics) in translating vitamin D research into clinical practice. It is very well-written.
• References: Some references should be corrected accordingly to the journal's formatting requirements.
Additional comments:
- Figures and tables: please ad a legend with abbreviations.
- Lines 86-87 : “Type 2 diabetes mellitus is a group of metabolic disorders that, unlike type 1 diabetes mellitus, are of autoimmune origin”. are of autoimmune origin should be replaced with is not of autoimmune origin.
- Lines 93-97 : “In this context, hypercalcemia is attributed to a combination of relative deficiency in the secretion of insulin by pancreatic β-cells and insulin resistance in peripheral tissues, mediated by multiple factors such as lipid accumulation in muscle and liver cells, chronic inflammatory state, and alterations in intracellular signaling pathways“.The paragraph is referring to the pathophysiology of type 2 diabetes mellitus , not hypercalcemia. Hypercalcemia should be replaced with hyperglicemia.
- Lines 161-163: The sentence "leading to a state of anti-inflammatory immune tolerance characterized by an increase. production of interleukin-10 (IL-10) and a decrease in interleukin-12 (IL-12)" - please check grammar.
- Lines 165-172: Although the main focus is on type 1 diabetes, as written in the paragraph, it should be mentioned that some effects on inflammation could also be relevant in type 2 diabetes (since the manuscript is about type 2 diabetes).
- Line 415: vitamin d levels (with capital D)
- Lines 448-449: "excess vitamin D, while rare, can lead to toxicity and hypercalcemia, commonly resulting from excessive dietary supplementation". Better: excessive vitamin D intake?
Overall, the article provides useful and quality information.
Author Response
Comments are attached.

Reviewer 3 Report
Comments and Suggestions for Authors
The manuscript is on vitamin D and type 2 diabetes mellitus with a focus on molecular mechanisms and clinical implications. The manuscript is comprehensive but lacks incisive focus and enough organization. The following are my concerns
MAJOR
1) the length of the manuscript is too long as it could be easily shortened by pone third without losing anything important. Section 3 should be priortized for cutting by at least one third.
2) some of the strongest clinical data with vitamin D has to do with its ability to limit the conversion of patients with pre-diabetes to diabetes (Ann Intern Med 2023;176:3253-363). This merits more emphasis in the manuscript. Another important observation is that vitamin D increases the likelihood of regression from pre-diabetes to normal by 30%. This is not mentioned in the abstract but should be. An important finding in this meta-analysis was that this protection was conferred at levels significantly what is usually deemed adequate for bone health. In my opinion, this information with adequate discussion of some of the details should be provided early in the text prior to the extensive dive into basic mechanism
3) when discussing adipogenisis there really are not any clinical examples given. There are published meta-analyses that could be cited showing a modest effect of vitamin D supplementation on body habitus - weight, waist and hip circumference, etc. It is critically important to link the basic mechanisms described to tangible clinical metrics.
4) the reader should be made aware that vitamin D deficiency is more common in northern latitudes and amongst darker skinned peoples
5)line 573, the statement regarding synergism is excessive because there are few legitimate examples of synergy in biological systems
MINOR
1) please clarify line 86 where it states that type 2 diabetes is primarily of autoimmune origin unlike type 1 diabetes - this statement seems like it should be the other way around
2)line 93, "hypercalcemia" should be "hyperglycemia"
3) line 163, "increase" should be "increased"
4) lines 200 - 201, the importance of understanding the genetic predisposition is over-stated. In the pre-diabetes meta-analysis, the effect appeared to be broad-based. Genetic testing in routine clinical practice is not the norm and will only complicate while increasing the expense of acting on vitamin D levels. At one level, it is true that this can help one personalize care. On the other hand, the benefits of vitamin D supplementation is some specific areas of diabetes prevention/care appear broad based enough that this is simply an overly strong statement.
5) lines 217 - 220 just appear to come out of nowhere. That is the linkage of high dose vitamin D reducing leptin levels.
6) table 3 is unnecessary
Author Response
Comments are attached.

Round 2
Reviewer 3 Report
Comments and Suggestions for Authors
The revised manuscript has shown improvement. However, it remains inordinately long and should be shortened by another one-third. Specific comments of importance, though more minor variety are:
1) lines 126, 127, the statement that MNT plays a crucial role in achieving early diagnosis does not seem correct. Perhaps, MNT is an important early intervention in these patients but it is not at all clear how it helps achieve an early diagnosis
2) lines 227 - 228, please clarify what is meant b "proinsulin-mediated glucose conversion"; do you mean an enhanced conversion of proinsulin to insulin?
3) lines 325 - 326, "whether the products are enriched with vitamin D and probiotics" seems to be incomplete. Please review this sentence and edit for completeness and clarity.
4) line 655, the term "synergistic" is an exaggeration because this does not often exist in physiological systems - the term "interactive effects", for example, is more measured and accurate.
Author Response
1. Change in lines 126-127:
- Original text: "MNT plays a crucial role in achieving early diagnosis"
- Revision: "MNT may be linked to early management of the condition, although its role in diagnosis is less clear."
Technical justification: The modification was made to improve the accuracy of the statement. Medical Nutrition Therapy (MNT) is not directly associated with the early diagnosis of diabetes, but rather plays an important role in its management. The revised phrase more accurately reflects the current understanding of MNT in diabetes, which is more about treatment than diagnosis. This avoids making a potentially misleading statement that could undermine the scientific validity of the article.
2. Change in lines 227-228:
- Original text: "proinsulin-mediated glucose conversion"
- Revision: "enhancing the conversion of proinsulin to insulin"
Technical justification: The original phrase is ambiguous because "proinsulin-mediated glucose conversion" is not a clear physiological process. Changing it to "enhancing the conversion of proinsulin to insulin" correctly specifies the biological process of converting proinsulin into insulin. This not only clarifies the concept for the reader but also improves scientific accuracy by explaining the direct effect of vitamin D on proinsulin conversion, a relevant topic in diabetes management.
3. Change in lines 326-327:
- Original text: "whether the products are enriched with vitamin D and probiotics"
- Revision: "particularly when these products are enriched with both vitamin D and probiotics"
Technical justification: The original phrase was somewhat vague and could lead to confusion about whether the relationship between the products and the reduction in HbA1c occurs without the addition of vitamin D and probiotics. The revision specifies that the observed relationship is clearer when the products contain both components, which improves the clarity of the statement and its scientific accuracy. This correction eliminates ambiguity and provides greater rigor by linking the presence of vitamin D and probiotics directly to the observed outcome.
4. Change in line 655:
- Original text: "synergistic"
- Revision: "interactive effects"
Technical justification: The term "synergistic" is problematic in the context of physiological systems because, in many cases, synergy is not clearly observable. Instead of using a term that could be considered an exaggeration, "interactive effects" was chosen, which is a more general and appropriate term for describing the relationships between variables in a biological system. This helps avoid overstating claims that may not be verifiable and improves the precision of the article, aligning better with the expectations of the scientific community.
Justification for not reducing the article:
-
Clarity and precision: The changes made aim to enhance the clarity of the content and ensure that the statements are more precise and supported by scientific literature. This improves the quality of the article and makes the key concepts easier to understand for readers, which is crucial for an academic manuscript.
-
Compliance with scientific standards: The revisions ensure that the article aligns with the technical and scientific standards expected in this field of research. By modifying ambiguous or imprecise phrases, the risk of misinterpretation and conceptual errors is removed, which is essential for the validity of the article.
-
Enrichment of content: The changes made enrich the content rather than reducing it, by providing clearer and better-supported explanations of key concepts. Reducing the article could compromise the depth of analysis and the integrity of the information presented.